# The current global perspective of the knowledge-attitude-behavior of the general public towards the corona virus disease -19 pandemic: Systematic review and meta-analysis on 67,143 participants

**Abdulhadi A. AlAmodi[1]☯, Khaled Al-Kattan[2], Mohammad Abrar Shareef[ID][3]☯ \***

1 Department of Epidemiology and Biostatistics, School of Public Health, Jackson State University, Jackson, Mississippi, United States of America, 2 Dean of college of Medicine, Alfaisal University, Riyadh, Saudi Arabia, 3 Department of Internal Medicine, Sebasticook Valley Hospital, Pittsfield, Maine, United States of America

☯ These authors contributed equally to this work.
\* mshareef@emhs.org

**Data Availability Statement:** All relevant data are within the manuscript and its Supporting information files.

## Abstract

### Background

Determining the success of infectious disease outbreak prevention is dependent mainly on public knowledge and compliance regarding the guidelines of precautionary behaviors and practices. While the current literature about the COVID-19 pandemic extensively addresses clinical and laboratory-based studies, a gap remains still present in terms of evaluating the general public knowledge and behaviors towards the COVID-19 pandemic. The aim of this review was to form a preliminary and contemporary understanding of the general public knowledge, attitude, and behaviors towards the COVID-19 pandemic globally.

### Methods

A systematic search was conducted in various databases until May 2020. Each study's characteristics including the sample size, region, and study type were examined individually. A meta-analysis with a random-effects model and pooled prevalence with 95% confidence interval (CI) of all evaluated outcomes such as adequate knowledge, positive feelings, worrisome about the COVID-19 pandemic, and practice were recorded and reported from each study. Parameters such as random distribution, blinding, incomplete outcome data, selective reporting, and other biases were utilized to assess the quality of each retrieved record. Both Begg's and Egger's tests were employed to evaluate symmetry of funnel plots for assessment of publication bias. The overall quality of evidence was evaluated using GRADEpro software.

### Results

A total of 26 studies with 67,143 participants were analyzed. The overall prevalence of knowledge, positive attitude, worrisome, and practice of precautionary measures were 0.87

**Funding:** The author(s) received no specific funding for this work.

**Competing interests:** The authors have declared that no competing interests exist.

(95%CI, 0.84–0.89), 0.85 (95%CI, 0.77–0.92), 0.71 (95%CI, 0.61–0.81), and 0.77 (95%CI, 0.70–0.83), respectively. Subgroup analysis demonstrated that social distancing was less practiced in Africa than other regions (p = 0.02), while knowledge of prevention of COVID-19 was reported higher in Asia (p = 0.001). Furthermore, people in developing countries had a higher prevalence of worrisome towards the COVID-19 pandemic with a p-value of less than 0.001. The quality of evidence was noted to be of low certainty in practice domain but moderate in the remaining outcomes.

## Conclusion

Assessing the public's risk perception and precautionary behaviors is essential in directing future policy and health population research regarding infection control and preventing new airborne disease outbreaks.

## Introduction

The novel coronavirus disease-2019 (COVID-19) pandemic represents an unprecedented crisis in the modern era resulting in deleterious consequences on public health, the economy, and healthcare systems [1–3]. On 31 December 2019, the World Health Organization (WHO) office in China was alerted to pneumonia cases of unknown origin in Wuhan City in the Hebei province [2]. By early January 2020, the Chinese government announced a new coronavirus that was later, on 11 February 2020, named the severe acute respiratory syndrome coronavirus 2 (SARS-CoV-2), which causes COVID-19. On 13 January 2020, the first COVID-19 case outside China was reported in Thailand [4]. On 30 January 2020, the WHO announced a global emergency, and by 11 March 2020, the WHO declared COVID-19 a global pandemic [5]. By the end of May 2020, there were approximately 6 million COVID-19 cases with 360,000 deaths worldwide [6].

Governments across the globe initiated, with various success rates, different responses involving all societal aspects to combat the spread of COVID-19. During the development of therapeutics and vaccines, the mainstay strategy to contain the spread of COVID-19 consists of following global and governmental health organizations' recommendations and self-isolation guidelines and social distancing. China was successfully able to halt the exponential increase of COVID-19 cases by entirely restricting the mobility of residents in and between cities. Similarly, Italy implemented a lockdown on a large part of the country and prevented public mass gatherings. The United States has gradually, with various degrees of restrictions based on the state, placed guidelines for its citizens two weeks after the first confirmed case. The United Kingdom, however, delayed the prevention of public congregations and closure of school following the recommendations of its scientific advisors [7].

Over the past two decades, infectious respiratory disease outbreaks repeatedly occurred, including the severe acute respiratory syndrome (SARS) in 2003, the swine flu in 2009, and the Middle East respiratory syndrome in 2012 [8]. Health organizations employ necessary standard measures to address new infectious disease outbreaks, such as identifying the pathogen's characteristics and dynamics, enhancing the capacity of diagnostics and screening, and development of therapeutics and vaccines [9]. Such measures are significant in determining the success of infectious disease outbreak prevention; however, they are also largely dependent on the public's compliance regarding the guidelines of precautionary behaviors and practices [9, 10].

Health behavior theories suggested that risk perception is central in determining individuals' precautionary behaviors. Risk communication forms the basis of risk perception, which promotes accurate knowledge enabling precautionary behaviors and practices [9, 10]. Taken together, compliance of the general public in following preventive measures plays a critical factor in reducing the widespread transmission of COVID-19. Therefore, the public's awareness is a fundamental element in the overall public health response to the COVID-19 pandemic.

While the current literature about the COVID-19 pandemic extensively addresses clinical and laboratory-based studies, a gap remains still present in terms of evaluating the general public knowledge and behaviors towards the COVID-19 pandemic. We conducted a systematic review and meta-analysis of the existing literature (as of this writing) [11–36] regarding the general public's knowledge, attitudes, and practices (KAP) towards the COVID-19 pandemic globally. To the best of our knowledge, this is the first-ever published comprehensive review on this topic.

## Materials and methods

### Search strategy and eligibility criteria

The systematic review was processed using the PRISMA (Prepared Items for Systematic Reviews and Meta-Analysis guidelines. The search strategy used a combination of the MeSH terms that include "COVID-19", "SARS-COV-2", "COVID", "knowledge", "attitude", and "practice" as illustrated in S1 Table. The search was performed in different databases consisting of Medline using PubMed, Cochrane Library, Science Direct, and Google Scholar. All retrieved records were screened for duplications using EndNote software, which were removed if found. The initial screening process included evaluating the title, and abstract. To determine the potentially eligible studies, we included studies of only the English language, any region worldwide, published or in print, and available full-text articles. Methodologically, we included only cross-sectional studies that reported outcomes of knowledge, attitudes and precautionary behaviors towards the COVID-19 pandemic among the general public. No restriction was applied in terms of sample size, study setting, data collection protocol, or study type. We excluded studies from healthcare providers, reports from children or high school students, studies reporting perception towards coronaviruses other than COVID-19, and studies that lacked reporting the measured outcomes.

### Data extraction

We extracted the following information: name of the first author, year of publication, study location (ie, country and region), sample size, study type, and reported outcomes. Outcomes were divided into 4 major domains: adequate knowledge, positive attitude, worrisome about COVID-19, and practice. The prevalence of each component under each domain was extracted from the included studies. For instance, in the knowledge domain, the correct response rates towards the clinical manifestation of COVID-19, prevention, transmission, identifying high-risk groups, and treatment were obtained. The prevalence of positive attitude towards COVID-19 in addition to the worrisome rate of acquiring COVID-19 was also obtained. Finally, the prevalence of handwashing practice, wearing a mask, and social distancing was also obtained.

### Quality of evidence and risk bias assessment

We independently evaluated the quality of retrieved records using the Cochrane's review guidelines for risk of bias assessment of cross-sectional studies. Evaluated items in each record

included random distribution, blinding, incomplete outcome data, selective reporting, and other bias. Studies were categorized as high risk of bias, low risk of bias, or unclear risk of bias using the abovementioned items. Studies that had an average scoring above three were designated as average quality [37].

The overall quality of evidence was assessed using the Grading of Recommendation, Assessment, Development, and Evaluation (GRADE). This tool examines various factors including the risk of bias, directness, consistency and precision of results in addition to publication bias. The GRADE certainty of each outcome may be high, moderate, low, or very low based on the aforementioned factors.

## Statistical analysis

The descriptive analysis was performed to report the characteristics of the included studies. The prevalence of appropriate knowledge and practice in each study was calculated by computing the average prevalence of components under each respective domain. The standard error of each study outcome was calculated by measuring the square root of the [reported prevalence multiplied by 1-prevalence and divided by sample size]. This was computed after ensuring that all outcomes met the requirement of $n \times p > 5$ and $n \times (1\text{-}p) > 5$, where n represents the sample size of each study, and p denotes the prevalence of a measured outcome.

The meta-analysis was processed via plotting the prevalence from each study and its weighted average; the latter was estimated by calculating standard error. The analysis used the inverse variance method, and the effect size was reported as the mean of the pooled prevalence with a 95% confidence interval. Heterogeneity was analyzed using the $I^2$ test, which demonstrates the proportion of variation among the studies that are not due to chance but to heterogeneity. A percentage of <50% was considered low, but if greater than 50%, the random effect model was used to summarize the results.

The subgroup analysis was performed in two steps. First, an analysis of the major domains was stratified by the type of country where each study was conducted and illustrated in forest plots. Next, each component under knowledge and practice domains was stratified by the study's regional location, and the outcome values are depicted in a table. Sensitivity analysis examines the difference in overall outcome results after removing each study and rerunning the analysis. Publication bias was assessed by generating funnel plots and examining its symmetry using Begg's and Egger's tests. The statistical analysis was performed using Review Manager version 5.3. Both Begg's and Egger's tests were utilized using MedCalc software. The GRADEpro GDT (Guideline Development Tool) software was utilized to evaluate the overall quality evidence. A p-value of less than 0.05 was considered statistically significant. The outcome data are presented as mean with a 95% confidence interval.

## Results

### Search results and descriptive characteristics

A total of 1383 articles were retrieved from a comprehensive search strategy in 4 different databases. When using the MeSH keywords in the Medline database, we noted a significant growth of literature since the beginning of 2020 (Fig 1). Most of the retrieved articles from the databases were removed for multiple reasons, most commonly due to topics, study subjects, and measured outcomes outside of this review's scope. The studies were collected from January 1st until May 20th. The final number of studies included 26 studies comprising 67,143 participants (Fig 2). A total of 18 studies were from developing countries, while 8 were from developed nations. All studies were cross-sectional and published as full articles or ahead of print in 2020

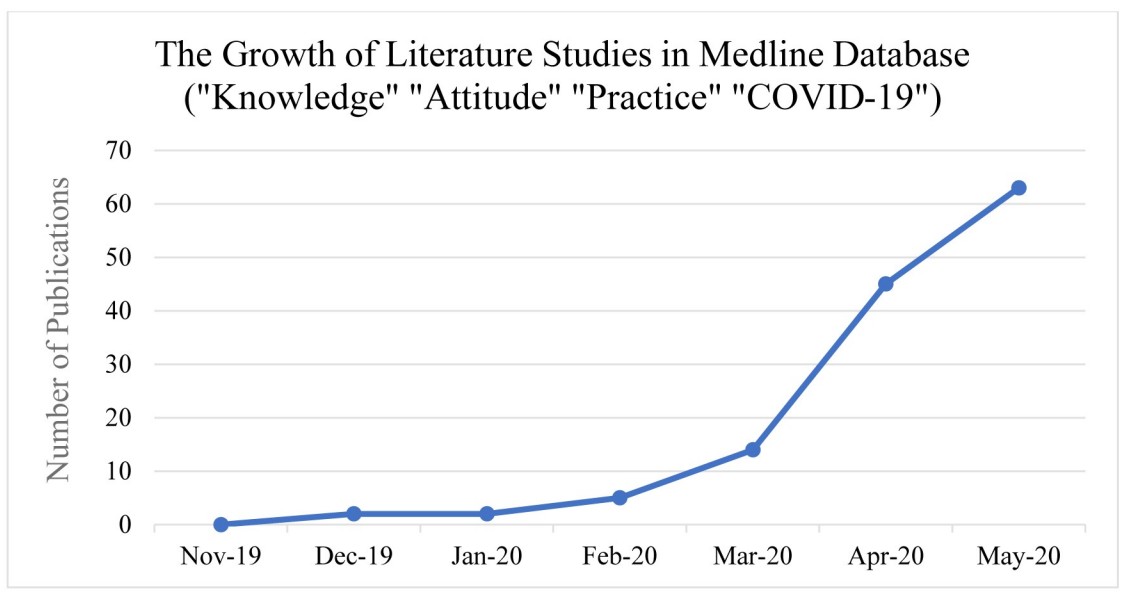

**Fig 1. The exponential growth of literature related to COVID-19 knowledge, attitudes, and practice among the public.**

(Table 1). Due to high heterogeneity with $I^2$ ranging from 82% to 100%, the random effect model was deployed for all group and subgroup analyses.

## Quality assessment outcomes

Table 2 displays the outcomes of the quality of assessment of all included studies. A total of 11 studies had a score of 4, while 10 studies had a score of 3, and 5 studies had a score of 2.

The GRADE scoring of each outcome revealed moderate level of evidence in knowledge, positive attitude and worrisome about COVID-19 domains. However, due to the presence of publication bias in the practice domain, the overall level evidence was noted to be low (Table 3).

## Prevalence of appropriate/adequate knowledge about COVID-19

The overall prevalence of knowledge about COVID-19 was 0.87 (95%CI, 0.84–0.89) (Table 4). Sensitivity analysis showed no significant difference in reported data when each study was excluded (Fig 3). Both Begg's and Egger's tests indicated statistically significant asymmetry of the funnel plot with p values of 0.60 and 0.10, respectively (Fig 4).

Stratification analysis did not demonstrate a statistically significant difference between the prevalence of adequate knowledge about COVID-19 between participants from developing and developed countries (p = 0.41; Fig 3). When examining the difference in the prevalence of different components under the knowledge domain between different regions, participants from Asia reported a higher rating in prevention knowledge than their peers in other regions with a p-value of 0.001 (Table 5).

## Attitude of study subjects towards COVID-19

A total of 9 studies with 33,944 participants have evaluated the positive attitude of study subjects toward COVID-19 and revealed an overall positive rating of 0.85 (95%CI, 0.77–0.92; (Table 4). Due to these studies' locations in developing countries, further stratification was not

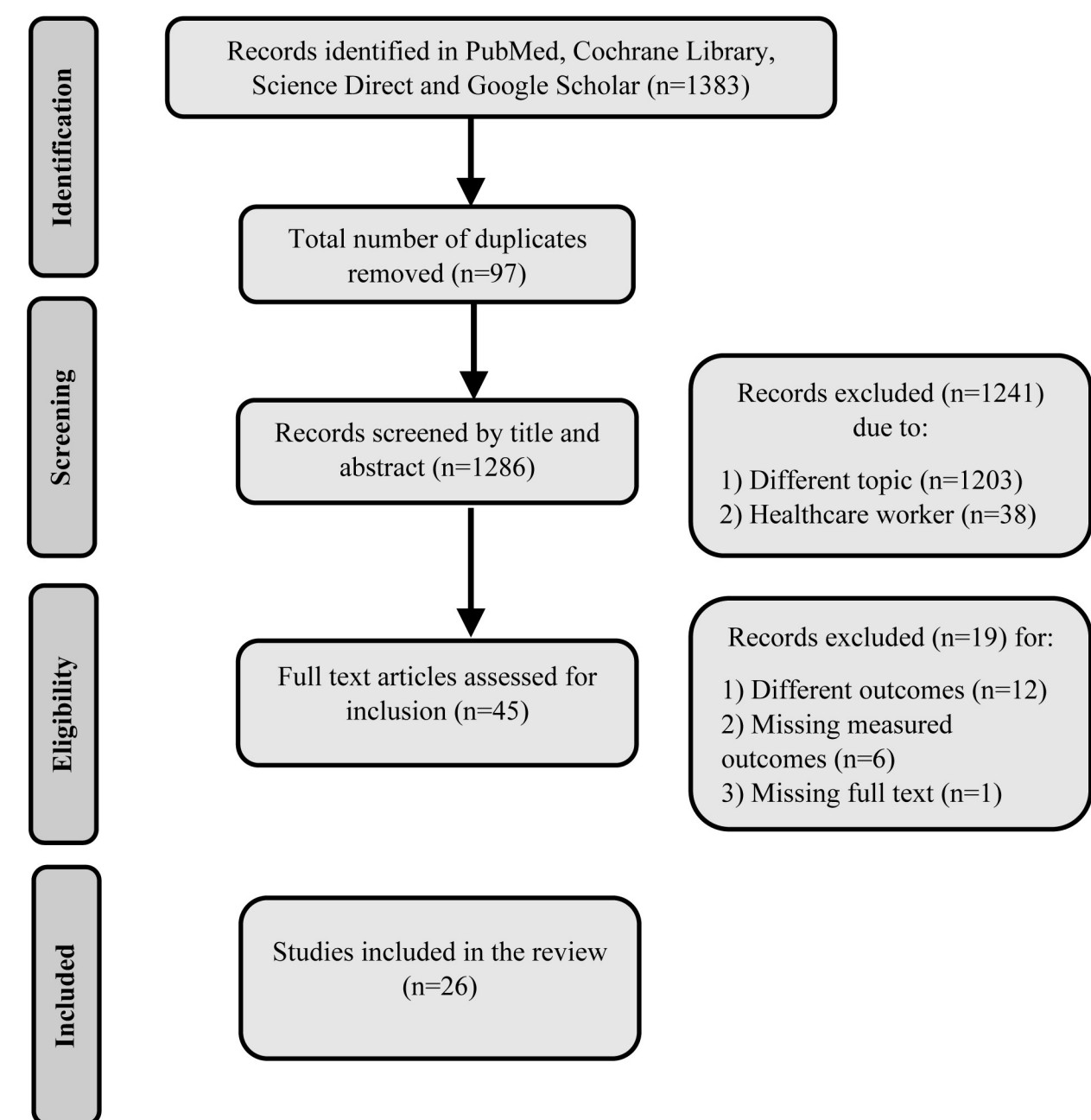

**Fig 2. The search strategy employing the PRISMA.** PRISMA = Preferred Reporting Items for Systematic Reviews and Meta-Analyses.

permissible (Fig 5). Sensitivity analysis showed no significant difference in the overall result after removing studies one at a time, indicating an overall reliable result. However, both Begg's and Egger's tests showed no statistically significant asymmetry of the funnel plot with p values of 0.75 and 0.91, respectively indicating low risk of publication bias (Fig 6).

On the contrary, studies reporting the prevalence of worrisome of its participants about COVID-19 demonstrated that around 71% of people were worried about contracting COVID-19 (Table 4). For instance, people from developing countries appeared to self-report a higher worrisome rate than those in developed countries with a p-value of less than 0.001 (Fig 7).

**Table 1. The characteristics of the included studies in the review.**

| N | Author | Year | Country | Sample size | Study type | Reported outcomes |
|---|--------|------|---------|-------------|-----------|-------------------|
| 1 | Abdelhafiz et al. [11] | 2020 | Egypt | 559 | Cross-sectional study | Knowledge |
| | | | | | | Attitude |
| | | | | | | Practice |
| 2 | Alzoubi et al. [12] | 2020 | Jordan | 592 | Cross-sectional study | Knowledge |
| | | | | | | Practice |
| 3 | Austrian et al. [13] | 2020 | Kenya | 2009 | Cross-sectional study | Knowledge |
| | | | | | | Attitude |
| 4 | Azlan et al. [14] | 2020 | Malaysia | 4850 | Cross-sectional study | Knowledge |
| | | | | | | Attitude |
| | | | | | | Practice |
| 5 | Chen et al. [15] | 2020 | China | 4061 | Cross-sectional study | Knowledge |
| | | | | | | Practice |
| 6 | Clements et al. [16] | 2020 | United States | 1070 | Cross-sectional study | Knowledge |
| | | | | | | Practice |
| 7 | Cowling et al. [17] | 2020 | Hong Kong | 3018 | Cross-sectional study | Attitude |
| | | | | | | Practice |
| 8 | Erfani et al. [18] | 2020 | Iran | 8591 | Cross-sectional study | Knowledge |
| | | | | | | Attitude |
| | | | | | | Practice |
| 9 | Geldsetzer et al. [19] | 2020 | UK & United States | 2988 (UK) | Cross-sectional study | Knowledge |
| | | | | 2986 (US) | | |
| 10 | Hayat et al. [20] | 2020 | Pakistan | 1257 | Cross-sectional study | Knowledge |
| | | | | | | Attitude |
| | | | | | | Practice |
| 11 | Keeling et al. [21] | 2020 | Ireland | 103 | Cross-sectional study | Knowledge |
| | | | | | | Attitude |
| 12 | Lima et al. [22] | 2020 | Brazil | 2259 | Cross-sectional study | Attitude |
| | | | | | | Practice |
| 13 | McFadden et al. [23] | 2020 | United States | 718 | Cross-sectional study | Practice |
| 14 | Misba et al. [24] | 2020 | Kashmir | 400 | Cross-sectional study | Knowledge |
| | | | | | | Attitude |
| | | | | | | Practice |
| 15 | Nwafor et al. [25] | 2020 | Nigeria | 284 | Cross-sectional study | Knowledge |
| 16 | Rios-Gonzalez et al. [26] | 2020 | Paraguay | 3141 | Cross-sectional study | Attitude |
| | | | | | | Practice |
| 17 | Roy et al. [27] | 2020 | India | 662 | Cross-sectional study | Knowledge |
| | | | | | | Attitude |
| | | | | | | Practice |
| 18 | Rugarabamu et al. [28] | 2020 | Tanzania | 400 | Cross-sectional study | Knowledge |
| | | | | | | Attitude |
| | | | | | | Practice |
| 19 | Salman et al. [29] | 2020 | Pakistan | 417 | Cross-sectional study | Knowledge |
| | | | | | | Attitude |
| | | | | | | Practice |
| 20 | Ssebuufu et al. [30] | 2020 | Uganda | 1763 | Cross-sectional study | Knowledge |
| | | | | | | Attitude |
| | | | | | | Practice |

*(Continued)*

**Table 1.** (Continued)

| N | Author | Year | Country | Sample size | Study type | Reported outcomes |
|---|--------|------|---------|-------------|------------|-------------------|
| 21 | Toan et al. [31] | 2020 | United States | 464 | Cross-sectional study | Knowledge |
| | | | | | | Attitude |
| | | | | | | Practice |
| 22 | Tomar et al. [32] | 2020 | India | 7978 | Cross-sectional study | Knowledge |
| | | | | | | Attitude |
| | | | | | | Practice |
| 23 | Wadood et al. [33] | 2020 | Bangladesh | 320 | Cross-sectional study | Knowledge |
| | | | | | | Attitude |
| 24 | Wolf et al. [34] | 2020 | United States | 630 | Cross-sectional study | Knowledge |
| | | | | | | Attitude |
| 25 | Zanin et al. [35] | 2020 | Italy | 8713 | Cross-sectional study | Attitude |
| 26 | Zhong et al. [36] | 2020 | China | 6910 | Cross-sectional study | Knowledge |
| | | | | | | Attitude |
| | | | | | | Practice |

**Table 2. The risk bias assessment of included records in this review.**

| Number | Author | Random distribution | Blinding | Incomplete outcome data | Selective reporting | Other bias | Total score |
|--------|--------|---------------------|----------|-------------------------|---------------------|------------|-------------|
| 1 | Abdelhafiz et al. [11] | L | U | L | L | L | 4 |
| 2 | Alzoubi et al. [12] | L | U | H | L | H | 2 |
| 3 | Austrian et al. [13] | L | U | H | L | L | 3 |
| 4 | Azlan et al. [14] | L | U | L | L | L | 4 |
| 5 | Chen et al. [15] | L | U | H | L | U | 2 |
| 6 | Clements et al. [16] | L | U | L | L | L | 4 |
| 7 | Cowling et al. [17] | L | U | U | L | L | 3 |
| 8 | Erfani et al. [18] | L | U | L | L | L | 4 |
| 9 | Geldsetzer et al. [19] | L | U | H | L | L | 3 |
| 10 | Hayat et al. [20] | L | U | L | L | H | 3 |
| 11 | Keeling et al. [21] | L | U | H | L | L | 3 |
| 12 | Lima et al. [22] | L | U | H | L | H | 2 |
| 13 | McFadden et al. [23] | L | U | L | L | L | 4 |
| 14 | Misba et al. [24] | L | U | L | L | L | 4 |
| 15 | Nwafor et al. [25] | L | U | H | L | L | 3 |
| 16 | Rios-Gonzalez et al. [26] | L | U | H | L | L | 3 |
| 17 | Roy et al. [27] | L | U | L | L | L | 4 |
| 18 | Rugarabamu et al. [28] | L | U | L | L | U | 3 |
| 19 | Salman et al. [29] | L | U | L | L | L | 4 |
| 20 | Ssebuufu et al. [30] | L | U | L | L | L | 4 |
| 21 | Toan et al. [31] | L | U | L | L | L | 4 |
| 22 | Tomar et al. [32] | L | U | L | L | H | 3 |
| 23 | Wadood et al. [33] | L | U | H | L | L | 3 |
| 24 | Wolf et al. [34] | L | U | H | L | U | 2 |
| 25 | Zanin et al. [35] | L | U | H | L | L | 2 |
| 26 | Zhong et al. [36] | L | U | L | L | L | 4 |

L = low; U = unclear; H = high.

**Table 3. The level of evidence of all measured outcomes using GRADE tool.**

| № of studies | Study design | Risk of bias | Inconsistency | Indirectness | Imprecision | Other considerations | № of participants | Prevalence (95% CI) | Certainty | Importance |
|---|---|---|---|---|---|---|---|---|---|---|
| | | | Certainty assessment | | | | | Effect | | |
| **Knowledge about COVID-19** | | | | | | | | | | |
| 21 | observational studies | not serious | serious [a] | not serious | not serious | none | 46308 | **0.87** (0.84 to 0.89) | ⊕⊕⊕◯ MODERATE | CRITICAL |
| **Positive Attitude towards COVID-19** | | | | | | | | | | |
| 9 | observational studies | not serious | serious [a] | not serious | not serious | none | 33944 | **0.85** (0.77 to 0.92) | ⊕⊕⊕◯ MODERATE | CRITICAL |
| **Worrisome about COVID-19** | | | | | | | | | | |
| 13 | observational studies | not serious | serious [a] | not serious | not serious | none | 29508 | **0.71** (0.61 to 0.81) | ⊕⊕⊕◯ MODERATE | CRITICAL |
| **Practice towards COVID-19** | | | | | | | | | | |
| 20 | observational studies | not serious | serious [a] | not serious | not serious | publication bias strongly suspected [b] | 57823 | **0.77** (0.7 to 0.83) | ⊕⊕◯◯ LOW | CRITICAL |

**CI**: Confidence interval. **COVID-19**: Corona virus disease -19.

[a]. Heterogeneity test with $I^2 > 95\%$.

[b]. Begg's test for publication bias (p<0.05)

Sensitivity analysis revealed adequacy and both Begg's (p = 0.27) and Egger's (p = 0.45) tests revealed no statistically significant asymmetry of the funnel plot (Fig 8).

## Prevalence of precautionary practice measures towards COVID-19

The use of overall practical precautions to limit the spread of COVID-19 was explored in most of the included studies with an average proportion of 0.77 (95%CI, 0.70–0.83) (Table 4). In general, no significant difference was noted in the utility of practical measures between participants from developed and developing countries (p = 0.28; Fig 9). Furthermore, sensitivity analysis showed no significant change in the outcome after removing each study and rerunning the model. However, publication bias was present using Begg's test with p value <0.01 indicating a statistically significant asymmetry of the funnel plot (Fig 10).

Concerning stratification analysis by regions, African participants self-reported a significantly lower social distancing rate than their peers from other regions (p = 0.02). Wearing masks in public was reported by 38% of North American participants, while 72% and 58% of those from Asia and Africa reported wearing a mask.

## Discussion

There is global consistency among the general public regarding the prevalence of two measured outcomes (knowledge and practice). The overall pooled prevalence in terms of an

**Table 4. Pooled prevalence of knowledge, attitude and practice towards COVID-19.**

| | Pooled prevalence | 95% confidence interval |
|---|---|---|
| Knowledge | 0.87 | 0.84–0.89 |
| Positive attitude | 0.85 | 0.77–0.92 |
| Worrisome | 0.71 | 0.61–0.81 |
| Practice | 0.77 | 0.70–0.83 |

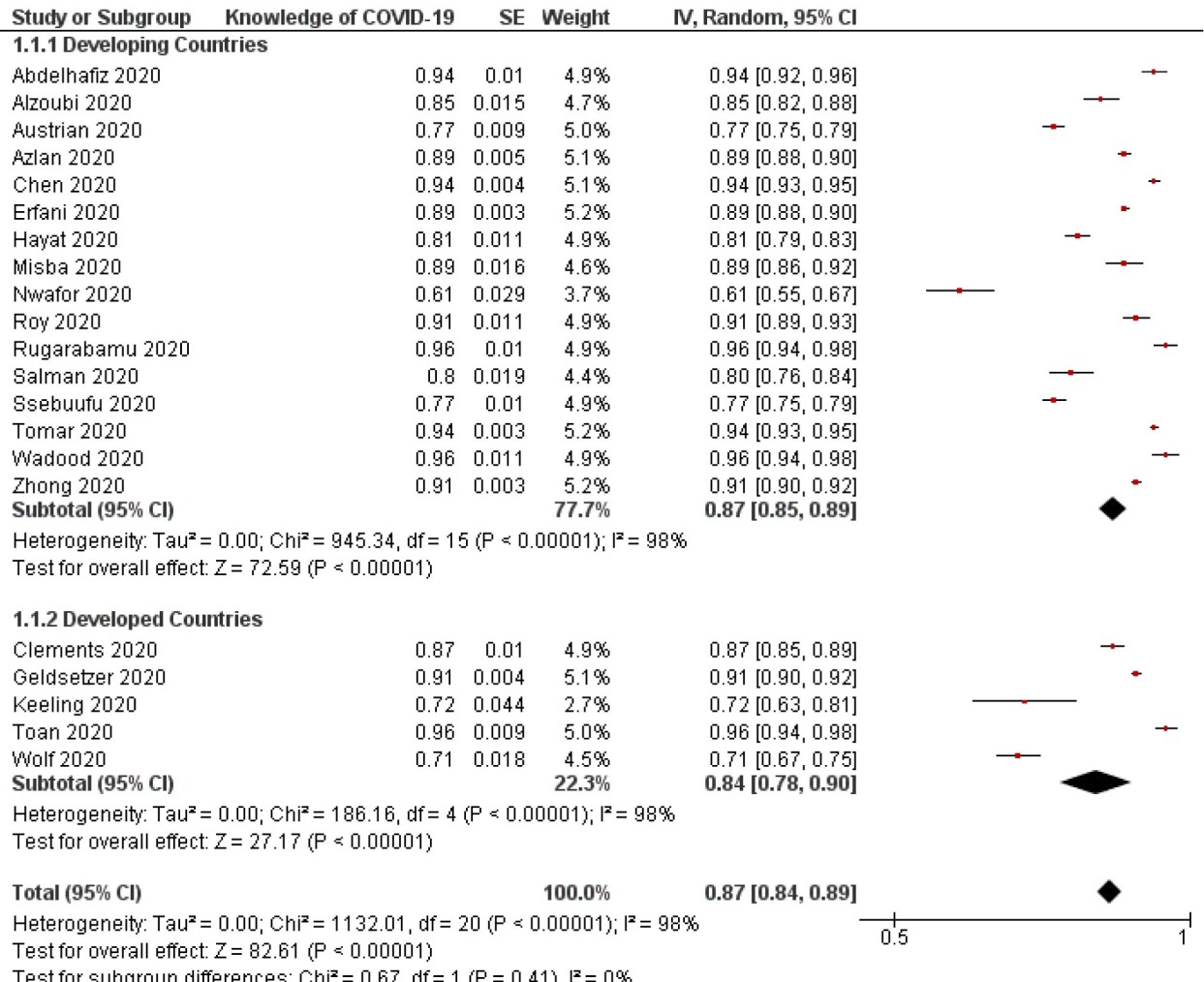

**Fig 3. Forest plot demonstrating prevalence of adequate knowledge about COVID-19 among study participants (n = 49,294).** Stratification analysis between developing and developed countries (p = 0.41). CI = confidence interval; IV = inverse variance; SE = standard error.

adequate knowledge level was 87%, with no statistically significant difference between developing (87%) and developed (84%) regions. Similarly, the overall pooled prevalence of precautionary behaviors and practices (social distancing, hand washing, and mask-wearing) was 77%. Even though not statistically significant, there was a slight difference between developing and developed regions regarding precautionary behaviors at 80% and 67%, respectively. In terms of attitudes, it was only assessed in developing countries, and 85% of the general public expressed positive feelings towards the implemented measures to contain the COVID-19 pandemic. Taken together, these findings demonstrate that the general public on a global level exhibit a favorable level of awareness and precautionary behavior during the COVID-19 pandemic.

The adequacy and consistency might be due to several factors. There has been global, large-scale effective communication about the COVID-19 pandemic between health organizations and the public. Because of recurrent and several worldwide outbreaks in the last two decades, the general public has become more aware and compliant in following precautionary behaviors during infectious disease outbreaks. Finally, the nature of this globalized pandemic in its

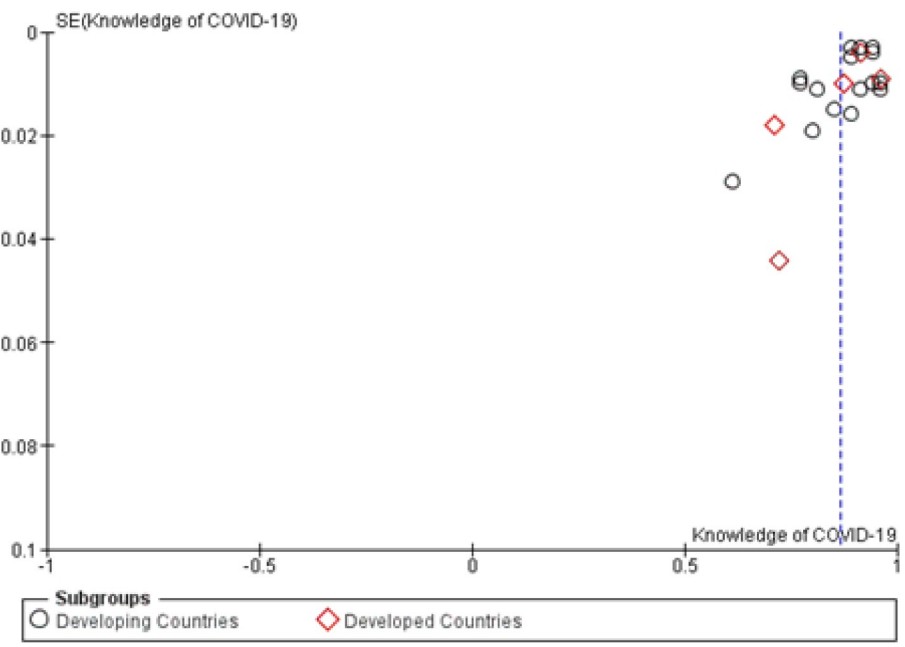

**Fig 4. Funnel plot of appropriate knowledge about COVID-19.** SE = standard error.

dynamics and viral kinetics, seriousness, and severity have made public communities more risk perceptive [8–10].

The current analysis highlighted several important differences between developed and developing regions regarding measured items belonging to each overarching domain. While the general public shows a consistent level of adequate knowledge across the globe, our analysis revealed that prevention knowledge was statistically significantly more substantial in Asia with a measured pooled prevalence of 95%. Given the timing of these studies, the first-impacted countries such as China and Thailand, and strict measures imposed by governments in these regions, the general population in these areas is expected to assume a great level of knowledge. On the other hand, countries in developing regions had a higher level of worrisome towards COVID-19 (79%; 95%CI, 69–88) than developed countries (58%; 95%CI, 54–62). Such findings could be attributed to the general public's perception and beliefs on the preparedness and response of healthcare systems' capacity and infrastructure in their countries [38, 39]. This effect could have been further compounded by proximal and distal mediators, including the mortality rate of COVID-19, experiences of immediate family members or friends, impact of quarantine, and more importantly, media and leadership influence and engagement [9, 39, 40]. In terms of social distancing practices, countries in Africa scored the lowest pooled prevalence with a statistical significance at 78% compared to Asia, North America, and South America. Such distinction is not as unpredictable as following WHO recommendations of physical distancing, and handwashing poses a major challenge in African countries due to poverty, overcrowding, and insufficiently prepared healthcare systems. The policy of physical distancing may yield a temporal economic value in high-income countries versus low-income countries, and in the latter, it could result in detrimental effects on laborers' income, especially in the absence of government policies directed towards aids reliefs of the population during this pandemic [41].

Previous reports related to recent airborne disease outbreaks demonstrated the significant role of the knowledge-attitude-behavior model in understanding the level of awareness among

**Table 5. Subgroup meta-analysis of knowledge and practice items using stratification by regions of reported studies.**

| | Regions Mean (95%CI) | | | | | |
| --- | --- | --- | --- | --- | --- | --- |
| | **Africa** | **Asia** | **Europe** | **North America** | **South America** | **P value** |
| **Knowledge** | **0.81 (0.71–0.91)** | **0.89 (0.87–0.91)** | **0.82 (0.63–1)** | **0.86 (0.80–0.93)** | - | **0.32** |
| $I^2$ | 99% | 97% | 95% | 98% | | |
| Wt. | 22% | 52% | 7% | 19% | | |
| 1) Clinical Presentation | 0.88 (0.76–1) | 0.93 (0.91–0.95) | 0.95 (0.87–1.0) | 0.90 (0.83–0.96) | - | 0.65 |
| $I^2$ | 99% | 99% | 86% | 99% | | |
| Wt. | 12% | 55% | 10% | 23% | | |
| 2) Prevention | 0.85 (0.77–0.94) | 0.95 (0.95–0.96) | - | 0.81 (0.73–0.90) | - | 0.001* |
| $I^2$ | 99% | 96% | | 98% | | |
| Wt. | 27% | 57% | | 16% | | |
| 3) Transmission | 0.94 (0.90–0.98) | 0.87 (0.80–0.95) | - | 0.83 (0.82–0.94) | - | 0.08 |
| $I^2$ | 84% | 100% | | 99% | | |
| Wt. | 15% | 69% | | 15% | | |
| 4) Identifying high risk group | 0.87 (0.70–1) | 0.81 (0.75–0.88) | 0.75 (0.32–1) | 0.92 (0.81–1) | - | 0.42 |
| $I^2$ | 100% | 100% | 99% | 99% | | |
| Wt. | 22% | 51% | 13% | 15% | | |
| 5) Treatment | 0.84 (0.62–1) | 0.92 (0.89–0.94) | - | - | - | 0.48 |
| $I^2$ | 100% | 98% | | | | |
| Wt. | 25% | 75% | | | | |
| **Practice** | **0.76 (0.64–0.88)** | **0.81 (0.77–0.85)** | - | **0.70 (0.46–0.94)** | **0.86 (0.76–0.96)** | **0.50** |
| $I^2$ | 98% | 100% | | 99% | 99% | |
| Wt. | 16% | 58% | | 16% | W11% | |
| 1) Hand washing | - | 0.87 (0.82–0.93) | - | 0.86 (0.71–1.0) | - | 0.85 |
| $I^2$ | | 100% | | 100% | | |
| Wt. | | 70% | | 30% | | |
| 2) Wearing mask | 0.58 (0.15–1.0) | 0.72 (0.66–0.79) | - | 0.38 (0–0.78) | - | 0.21 |
| $I^2$ | 100% | 100% | | 100% | | |
| Wt. | 12% | 69% | | 19% | | |
| 3) Social distancing | 0.78 (0.70–0.87) | 0.86 (0.81–0.90) | - | 0.87 (0.73–1) | 0.90 (0.88–0.92) | 0.02* |
| $I^2$ | 95% | 100% | | 99% | 82% | |
| Wt. | 17% | 53% | | 18% | 12% | |

the public towards an emerging outbreak and, hence, compliance towards infection control and prevention measures. The current expansion of technology and social media engagement mediated the appearance of many incredible resources spreading information and misinformation about health-related issues. In 2014, a study conducted by Jalloh demonstrated that a great proportion of the public had a misconception regarding the Ebola mode of transmission and prevention strategies [42]. Similarly, studies related to SARS and Zika outbreaks evaluated the extent of public compliance and the response toward mitigation activities [42]. A large-scale investigation included 13 surveys demonstrating that understanding the public's current knowledge, attitudes, and precautionary behaviors would facilitate public health officials and medical doctors' role in developing communication redresses. In the same study, only half of the surveyed individuals knew that there was no effective treatment for SARS [43]. A study conducted by Burg et al. during the SARS outbreak demonstrated that while the general public had a great level of awareness, precautionary measures were not necessarily implemented [44].

The current review expands on contemporary evidence related to the general public's KAP/ behaviors towards the airborne diseases in general and COVID-19 on a global level. Reviewing

| Study or Subgroup | Positive Attitude | SE | Weight | IV, Random, 95% CI | |
|---|---|---|---|---|---|
| Azlan 2020 | 0.83 | 0.005 | 11.2% | 0.83 [0.82, 0.84] | |
| Erfani 2020 | 0.68 | 0.005 | 11.2% | 0.68 [0.67, 0.69] | |
| Hayat 2020 | 0.74 | 0.012 | 11.1% | 0.74 [0.72, 0.76] | |
| Misba 2020 | 0.91 | 0.014 | 11.0% | 0.91 [0.88, 0.94] | |
| Rios-Gonzalez 2020 | 0.86 | 0.006 | 11.2% | 0.86 [0.85, 0.87] | |
| Rugarabamu 2020 | 0.96 | 0.01 | 11.1% | 0.96 [0.94, 0.98] | |
| Salman 2020 | 0.75 | 0.021 | 10.8% | 0.75 [0.71, 0.79] | |
| Tomar 2020 | 0.98 | 0.001 | 11.2% | 0.98 [0.98, 0.98] | |
| Zhong 2020 | 0.91 | 0.003 | 11.2% | 0.91 [0.90, 0.92] | |
| **Total (95% CI)** | | | **100.0%** | **0.85 [0.77, 0.92]** | |

Heterogeneity: Tau² = 0.01; Chi² = 5177.09, df = 8 (P < 0.00001); I² = 100%
Test for overall effect: Z = 22.95 (P < 0.00001)

**Fig 5. Forest plot illustrating prevalence of positive attitude towards COVID-19 among study subjects (n = 33,944).** CI = confidence interval; IV = inverse variance; SE = standard error.

all potentially available studies about the public, including over 67,000 participants, speaks to this review's strength. Further, we performed a subgroup analysis to illuminate differences based on geographical regions. Our study is the first to report a large-scale qualitative and quantitative-based review of COVID-19 perception among the general public. Besides, this review provides implications for future policy modifications and future research directions. Our study offers a new insight for policy makers in public health services. Efforts should be directed to consistently educate the public about this growing pandemic. More strict measures and policies should be highlighting the impact of physical separation, national mask mandate, and hand washing. Policy makers in government as well as the department of health shall

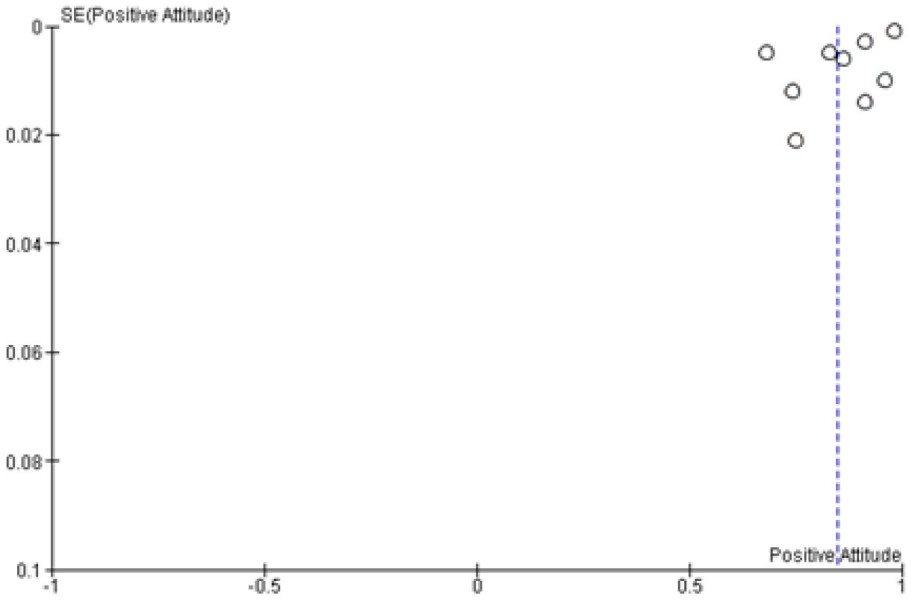

**Fig 6. Funnel plot examining the publication bias of prevalence of positive attitude towards COVID-19.** SE = standard error.

| Study or Subgroup | Worrisome about COVID-19 | SE | Weight | IV, Random, 95% CI | |
|---|---|---|---|---|---|
| **1.1.1 Developing Countries** | | | | | |
| Abdelhafiz 2020 | 0.86 | 0.015 | 7.7% | 0.86 [0.83, 0.89] | |
| Austrian 2020 | 0.68 | 0.01 | 7.7% | 0.68 [0.66, 0.70] | |
| Erfani 2020 | 0.78 | 0.004 | 7.8% | 0.78 [0.77, 0.79] | |
| Lima 2020 | 0.61 | 0.01 | 7.7% | 0.61 [0.59, 0.63] | |
| Roy 2020 | 0.72 | 0.017 | 7.7% | 0.72 [0.69, 0.75] | |
| Salman 2020 | 0.72 | 0.022 | 7.7% | 0.72 [0.68, 0.76] | |
| Ssebuufu 2020 | 0.97 | 0.004 | 7.8% | 0.97 [0.96, 0.98] | |
| Wadood 2020 | 0.95 | 0.012 | 7.7% | 0.95 [0.93, 0.97] | |
| **Subtotal (95% CI)** | | | **61.9%** | **0.79 [0.69, 0.88]** | |

Heterogeneity: Tau² = 0.02; Chi² = 2228.91, df = 7 (P < 0.00001); I² = 100%
Test for overall effect: Z = 15.72 (P < 0.00001)

| | | | | | |
|---|---|---|---|---|---|
| **1.1.2 Developed Countries** | | | | | |
| Cowling 2020 | 0.51 | 0.009 | 7.8% | 0.51 [0.49, 0.53] | |
| Keeling 2020 | 0.63 | 0.048 | 7.3% | 0.63 [0.54, 0.72] | |
| Toan 2020 | 0.62 | 0.023 | 7.7% | 0.62 [0.57, 0.67] | |
| Wolf 2020 | 0.63 | 0.019 | 7.7% | 0.63 [0.59, 0.67] | |
| Zanin 2020 | 0.55 | 0.005 | 7.8% | 0.55 [0.54, 0.56] | |
| **Subtotal (95% CI)** | | | **38.1%** | **0.58 [0.54, 0.62]** | |

Heterogeneity: Tau² = 0.00; Chi² = 49.33, df = 4 (P < 0.00001); I² = 92%
Test for overall effect: Z = 27.95 (P < 0.00001)

| | | | | | |
|---|---|---|---|---|---|
| **Total (95% CI)** | | | **100.0%** | **0.71 [0.61, 0.81]** | |

Heterogeneity: Tau² = 0.03; Chi² = 5991.45, df = 12 (P < 0.00001); I² = 100%
Test for overall effect: Z = 13.76 (P < 0.00001)
Test for subgroup differences: Chi² = 14.66, df = 1 (P = 0.0001), I² = 93.2%

**Fig 7. Forest plot depicting the difference in the rate of worrisome about COVID-19 between studies from developing and developed countries (n = 29,508, p<0.001).** CI = confidence interval; IV = inverse variance; SE = standard error.

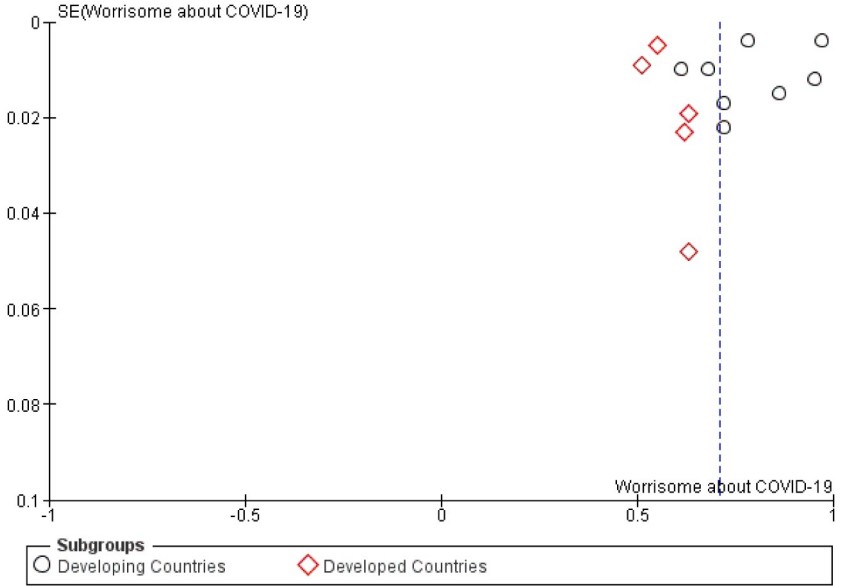

**Fig 8. Funnel plot demonstrating asymmetric distribution of self-reported ratings of worrisome about COVID-19.** SE = standard error.

| Study or Subgroup | Practice towards COVID-19 | SE | Weight | IV, Random, 95% CI |
|---|---|---|---|---|
| **1.1.1 Developing Countries** | | | | |
| Abdelhafiz 2020 | 0.65 | 0.02 | 4.9% | 0.65 [0.61, 0.69] |
| Alzoubi 2020 | 0.82 | 0.016 | 5.0% | 0.82 [0.79, 0.85] |
| Azlan 2020 | 0.85 | 0.005 | 5.0% | 0.85 [0.84, 0.86] |
| Chen 2020 | 0.93 | 0.004 | 5.0% | 0.93 [0.92, 0.94] |
| Erfani 2020 | 0.85 | 0.004 | 5.0% | 0.85 [0.84, 0.86] |
| Hayat 2020 | 0.85 | 0.01 | 5.0% | 0.85 [0.83, 0.87] |
| Lima 2020 | 0.91 | 0.006 | 5.0% | 0.91 [0.90, 0.92] |
| Misba 2020 | 0.5 | 0.025 | 4.9% | 0.50 [0.45, 0.55] |
| Rios-Gonzalez 2020 | 0.81 | 0.007 | 5.0% | 0.81 [0.80, 0.82] |
| Roy 2020 | 0.68 | 0.018 | 5.0% | 0.68 [0.64, 0.72] |
| Rugarabamu 2020 | 0.79 | 0.021 | 4.9% | 0.79 [0.75, 0.83] |
| Salman 2020 | 0.54 | 0.024 | 4.9% | 0.54 [0.49, 0.59] |
| Ssebuufu 2020 | 0.85 | 0.008 | 5.0% | 0.85 [0.83, 0.87] |
| Tomar 2020 | 0.95 | 0.002 | 5.0% | 0.95 [0.95, 0.95] |
| Zhong 2020 | 0.97 | 0.002 | 5.0% | 0.97 [0.97, 0.97] |
| **Subtotal (95% CI)** | | | 75.0% | 0.80 [0.77, 0.84] |
| Heterogeneity: Tau² = 0.00; Chi² = 2661.02, df = 14 (P < 0.00001); I² = 99% | | | | |
| Test for overall effect: Z = 44.83 (P < 0.00001) | | | | |
| | | | | |
| **1.1.2 Developed Countries** | | | | |
| Clements 2020 | 0.52 | 0.015 | 5.0% | 0.52 [0.49, 0.55] |
| Cowling 2020 | 0.85 | 0.007 | 5.0% | 0.85 [0.84, 0.86] |
| McFadden 2020 | 0.89 | 0.012 | 5.0% | 0.89 [0.87, 0.91] |
| Toan 2020 | 0.69 | 0.021 | 4.9% | 0.69 [0.65, 0.73] |
| Zanin 2020 | 0.42 | 0.005 | 5.0% | 0.42 [0.41, 0.43] |
| **Subtotal (95% CI)** | | | 25.0% | 0.67 [0.44, 0.91] |
| Heterogeneity: Tau² = 0.07; Chi² = 3201.60, df = 4 (P < 0.00001); I² = 100% | | | | |
| Test for overall effect: Z = 5.70 (P < 0.00001) | | | | |
| | | | | |
| **Total (95% CI)** | | | 100.0% | 0.77 [0.70, 0.83] |
| Heterogeneity: Tau² = 0.02; Chi² = 13239.82, df = 19 (P < 0.00001); I² = 100% | | | | |
| Test for overall effect: Z = 24.14 (P < 0.00001) | | | | |
| Test for subgroup differences: Chi² = 1.15, df = 1 (P = 0.28), I² = 12.9% | | | | |

**Fig 9. Forest plot demonstrating the prevalence of using practical measures against COVID-19.** Stratification analysis between developing and developed countries (p = 0.28). CI = confidence interval; IV = inverse variance; SE = standard error.

provide resources that will ultimately reduce the acquisition and transmission of COVID-19. A gap between first and second surge of CVOID-19 spread with absences of strict policies will create a more deleterious impact. Policy makers should utilize all available venues to spread credible information about the dynamics, updates, and seriousness of this novel coronavirus.

Our findings should be interpreted with caution as there are several critical limitations as follows: 1) these findings were restricted to the early duration of the outbreak, 2) the element of bias cannot be eliminated as participants responded subjectively, 3) studies with different qualities and variable instrumental tools were included, and 4) we included only those records available in English. Therefore, the publication bias was augmented due to heterogeneity resulting from variation in the sample size, methodological differences, inconsistent quality outcome of various studies, different regions, and sociodemographic factors. Our conclusions represent a preliminary trend that may ultimately change over time as the number of studies increases, the sample size is larger, less heterogeneity, and more rigorous methodological protocols."

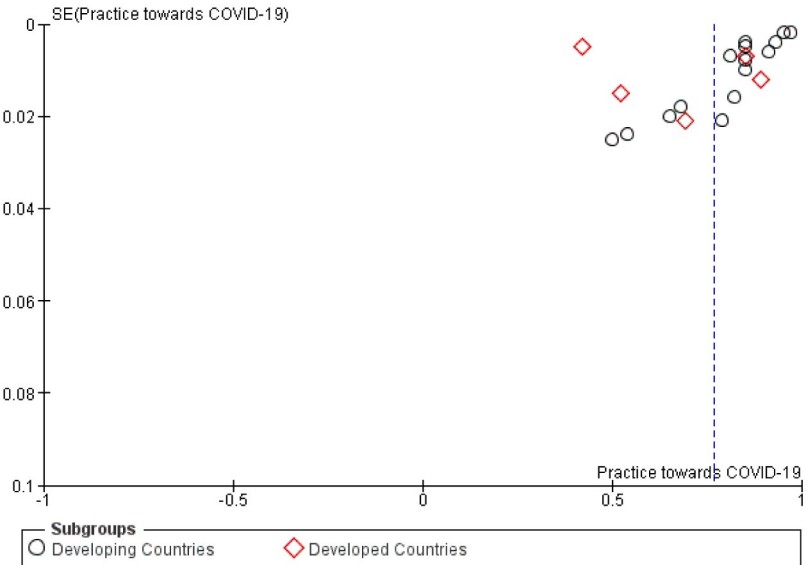

**Fig 10. Funnel plot evaluating publication bias of studies examining the use of practical measures against COVID-19.** SE = standard error.

In conclusion, our review shows a consistent contemporary global perspective among the public towards the COVID-19 pandemic. Regional differences to COVID-19 precautions included greater knowledge of the disease in Asia, physical distancing was less practiced in Africa, and more worrisome was expressed in developing countries compared to developed ones. Therefore, applying policies and increasing awareness will ultimately modify the general public knowledge, attitudes and practice towards the current pandemic. Future strategies should seek to improve public risk perception towards COVID-19 (especially after social distancing deactivation) and improve accessibility and availability of credible information. Additionally, future strategies should facilitate the investigation and characterization of under-represented minority groups' opinions and those living in rural areas.

## Supporting information

**S1 Table. A sample of search strategy in Medline database using MeSH keywords.**
(DOCX)

**S2 Table. PRISMA checklist.**
(DOC)

**S1 File. Dataset of extracted prevalence from reported studies.**
(XLSX)

## Author Contributions

**Conceptualization:** Abdulhadi A. AlAmodi, Mohammad Abrar Shareef.

**Data curation:** Abdulhadi A. AlAmodi, Mohammad Abrar Shareef.

**Formal analysis:** Abdulhadi A. AlAmodi, Mohammad Abrar Shareef.

**Investigation:** Abdulhadi A. AlAmodi, Khaled Al-Kattan.

**Methodology:** Abdulhadi A. AlAmodi, Khaled Al-Kattan, Mohammad Abrar Shareef.

**Project administration:** Mohammad Abrar Shareef.

**Supervision:** Mohammad Abrar Shareef.

**Validation:** Abdulhadi A. AlAmodi, Khaled Al-Kattan, Mohammad Abrar Shareef.

**Visualization:** Abdulhadi A. AlAmodi, Khaled Al-Kattan.

**Writing – original draft:** Abdulhadi A. AlAmodi, Khaled Al-Kattan, Mohammad Abrar Shareef.

**Writing – review & editing:** Abdulhadi A. AlAmodi, Mohammad Abrar Shareef.

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
