## [Decision Letter · Decision Letter 0]

24 Sep 2020

PONE-D-20-19020

The Current Global Perspective of the General Public towards the COVID-19 Pandemic Using the Knowledge-Attitude-Behavior Model: Systematic Review and Meta-Analysis on 67,143 Participants.

PLOS ONE

Dear Dr. Shareef,

Thank you for submitting your manuscript to PLOS ONE. After careful consideration, we feel that it has merit but does not fully meet PLOS ONE’s publication criteria as it currently stands. Therefore, we invite you to submit a revised version of the manuscript that addresses the points raised during the review process.

Thanks for submission to Plos One. Your manuscript has been reviewed by experts and found it interesting. However, there are few concerns related to the methodology of review including quality assessment and results presentations. These concerns are needed to be addressed before considering the draft for publication.

We look forward to receiving your revised manuscript.

Kind regards,

Tauqeer Hussain Mallhi, Ph.D

Academic Editor

PLOS ONE

Additional Editor Comments:

Thanks for submission to Plos One. Your manuscript has been reviewed by experts and found it interesting. However, there are few concerns related to the methodology of review including quality assessment and results presentations. These concerns are needed to be addressed before considering the draft for publication.

Journal Requirements:

2. Please provide a brief discussion on the effect of publication bias on your conclusions.

3. Please confirm that you have included all items recommended in the PRISMA checklist including the full electronic boolean search strategy used to identify studies with all search terms and limits for at least one database. Please attach this as supplementary file.

4. Please include the specific publication date range that was used to search the databases for literature.

Reviewers' comments:

Reviewer's Responses to Questions

**Comments to the Author**

1. Is the manuscript technically sound, and do the data support the conclusions?

Reviewer #1: Yes

Reviewer #2: Yes

Reviewer #3: Yes

2. Has the statistical analysis been performed appropriately and rigorously? 

Reviewer #1: Yes

Reviewer #2: Yes

Reviewer #3: Yes

3. Have the authors made all data underlying the findings in their manuscript fully available?

Reviewer #1: No

Reviewer #2: Yes

Reviewer #3: Yes

4. Is the manuscript presented in an intelligible fashion and written in standard English?

Reviewer #1: Yes

Reviewer #2: Yes

Reviewer #3: No

5. Review Comments to the Author

Reviewer #1: The study was a systematic review and meta-analysis on The Current Global Perspective of the General Public towards the COVID-19 Pandemic Using the Knowledge-Attitude-Behavior Model: Systematic Review and Meta-Analysis on 67,143 Participants

The analysis was rigorous and revealed the pattern of knowledge attitude and precautionary practice globally

Among such revelations is the worrisome behavior of developing countries compared to the developed world.

Another is that, precautionary measures were similar between developed and developing countries.

However, the authors should attach an excel file of the minimum data used in the analysis

It should be accepted for publication

Reviewer #2: It is a very relevant manuscript at present situation. There are just a few suggestions:

i) There are some grammatical and topographical errors detected throughout the manuscript and a professional editing is advised

ii) In the methods section the eligibility criteria can be further described applying stage 1 and stage 2 criteria. Stage 2 focusing more on the methodology of the eligibility criteria in terms of selection of research papers selected for the review to address the research question

iii) The discussion section can be expanded with suggesting more policy recommendations that the results of the review are suggesting

Reviewer #3: This review is a significant and timely upmost required to be conducted.

However, the general suggestions and comments should be incorporated well.

There is a language editorial problems. Eg tenses, spelling error (eg Review manager), grammar usage

I personally suggest the review title is required to be reshuffled as:

Title: The current global perspective of the knowledge-attitude-behavior of the general public towards the corona virus disease -19 Pandemic: Systematic Review and Meta-Analysis

Abbreviation shouldn’t be appeared in the title.

Authors information: adhere to Plos one guideline

Abstract

Research gap was not indicated

Objective the review was not well stated

Materials and Methods

Methods of data extraction and effect size measures were not specified.

Quality of evidence assessment was not explained.

Results

The finding the review was not well written up.

Introduction

Back ground

Research gap was not explained.

Materials and methods

What are your review research development?

Search strategy

The authors didn’t use MeSH terms which was inclusive

What did the authors use to avoid duplication of records?

Data extraction

Outcomes assessment was not explained

Quality and risk bias assessment

When the 2 author reviewers faced disagreement, how did you resolve it?

How could blinding be an evaluation item for quality assessment as your studies included was cross sectional study design?

Wht did you do if certain studies didn’t meet methodological criteria?

Statistical analysis

How did you express you effect size?

How did you assess publication bias more objectively? Explain it by using statistical tests

Quality of evidence assessment for major outcomes was not explained? Use Grade pro soft ware

Results

The pooled prevalence of each outcome along with CI should be specified separately in a logical order

Discussion

The finding was not compared and contrasted.

The possible reason of the differences/similarities should be stated

Conclusion

It should be made in line to the findings.

6. PLOS authors have the option to publish the peer review history of their article (what does this mean?). If published, this will include your full peer review and any attached files.

Reviewer #1: **Yes: **Abesig Julius

Reviewer #2: No

Reviewer #3: No

---

## [Author Response · Author response to Decision Letter 0]

15 Nov 2020

Editor’s Comments:

Point#1:

Authors’ Response:

Thank you for your note. We made sure the manuscript meets the requirements of PLOSONE.

Point#2:

Please provide a brief discussion on the effect of publication bias on your conclusions.

Authors’ Response:

 Thank you for your input.

The following segment was added in the limitation part of the discussion section of the paper. 

“Therefore, the publication bias was augmented due to heterogeneity resulting from variation in the sample size, methodological differences, inconsistent quality outcome of various studies, different regions, and sociodemographic factors. Our conclusions represent a preliminary trend that may ultimately change over time as the number of studies increases, the sample size is larger, less heterogeneity, and more rigorous methodological protocols.”

Point#3:

Please confirm that you have included all items recommended in the PRISMA checklist including the full electronic boolean search strategy used to identify studies with all search terms and limits for at least one database. Please attach this as a supplementary file.

Authors’ Response:

Thank you for this important note. We confirm that we included two supplementary tables. One for the PRISMA checklist and the second document explains the keywords used for search strategy in Medline database. 

Point#4:

Please include the specific publication date range that was used to search the databases for literature.

Authors’ Response: 

The following paraphrase was added in the results’ section of the manuscript:

“The studies were collected from January 1st until May 20th”. 

Point#5:

Please include captions for your Supporting Information files at the end of your manuscript, and update any in-text citations to match accordingly. Please see our Supporting Information guidelines for more information: http://journals.plos.org/plosone/s/supporting-information.

Authors’ Response: 

Thank you for this comment as well as the abovementioned comments. All adjusted according to per your recommendations. Supporting information files captions and files was added. 

Reviewers' Comments to the Authors:

Please use the space provided to explain your answers to the questions above. You may also include additional comments for the author, including concerns about dual publication, research ethics, or publication ethics. (Please upload your review as an attachment if it exceeds 20,000 characters).

Comments of Reviewer#1:

Point#1:

The study was a systematic review and meta-analysis on The Current Global Perspective of the General Public towards the COVID-19 Pandemic Using the Knowledge-Attitude-Behavior Model: Systematic Review and Meta-Analysis on 67,143 Participants. The analysis was rigorous and revealed the pattern of knowledge attitude and precautionary practice globally. Among such revelations is the worrisome behavior of developing countries compared to the developed world. Another is that precautionary measures were similar between developed and developing countries. However, the authors should attach an excel file of the minimum data used in the analysis. It should be accepted for publication.

Authors’ Response: 

Thank you for these highlights. We believe it will be the first comprehensive report that will provide a preliminary finding on a large scale as well as motivating more future investigations of similar types in the future. The raw data file was added as a supplementary file for your reference. 

Comments of Reviewer#2:

 It is a very relevant manuscript at present situation. There are just a few suggestions:

Point#1:

There are some grammatical and topographical errors detected throughout the manuscript and a professional editing is advised. 

Authors’ Response: 

Thank you for your note. We consulted with a professional medical writing service that corrected the mechanical errors and performed linguistic editing in the manuscript. Corrections are highlighted throughout the manuscript. 

Point#2:

In the methods section the eligibility criteria can be further described applying stage 1 and stage 2 criteria. Stage 2 focusing more on the methodology of the eligibility criteria in terms of selection of research papers selected for the review to address the research question.

Authors’ Response: 

Thank you for your valuable input. We added the following paragraph in the Methods section (search strategy and eligibility criteria): 

“The initial screening process included evaluating the title and abstract. To determine the potentially eligible studies, we included studies of only the English language, any region worldwide, published or in print, and available full-text articles. Methodologically, we included only cross-sectional studies that reported outcomes of knowledge, attitudes and precautionary behaviors towards the COVID-19 pandemic among the general public. No restriction was applied in terms of sample size, study setting, data collection protocol, or study type.” 

Point#3:

The discussion section can be expanded by suggesting more policy recommendations that the results of the review are suggesting.

Authors’ Response: 

Thank you for this comment. We added the following in the discussion. 

“Our study offers a new insight for policy makers in public health services. Efforts should be directed to consistently educate the public about this growing pandemic. More strict measures and policies should be highlighting the impact of physical separation, national mask mandate, and hand washing. Policy makers in government as well as the department of health shall provide resources that will ultimately reduce the acquisition and transmission of COVID-19. A gap between first and second surge of CVOID-19 spread with absences of strict policies will create a more deleterious impact. Policy makers should utilize all available venues to spread credible information about the dynamics, updates, and seriousness of this novel coronavirus.” 

Comments of Reviewer#3:

This review is a significant and timely upmost required to be conducted. However, the general suggestions and comments should be incorporated well.

Point#1:

There are language editorial problems. Eg tenses, spelling error (eg Review manager), grammar usage.

Authors’ Response: 

Thank you for your comment. We consulted with a professional medical writing service that corrected the mechanical errors and performed linguistic editing in the manuscript. Corrections are highlighted throughout the manuscript.

Point#2:

I personally suggest the review title is required to be reshuffled as:

Title: The current global perspective of the knowledge-attitude-behavior of the general public towards the corona virus disease -19 Pandemic: Systematic Review and Meta-Analysis. Abbreviation shouldn’t be appeared in the title.

Authors’ Response: 

Thank you for your suggestion. We changed the title according to your recommendation and added the number of participants to emphasize the strength of the study. We removed the abbreviation. The following change was incorporated: 

“The current global perspective of the knowledge-attitude-behavior of the general public towards the coronavirus disease -19 Pandemic: Systematic review and meta-analysis on 67,143 participants.”

Point#3:

Authors information: adhere to Plos one guideline.

Authors’ Response: 

Changed per PLOS one guideline. 

Point#3:

Abstract: 

Research gap was not indicated

Authors’ Response: 

Thank you for your valuable note. We incorporated the following in the abstract’s background. 

“While the current literature about the COVID-19 pandemic extensively addresses clinical and laboratory-based studies, a gap remains still present in terms of evaluating the general public knowledge and behaviors towards the COVID-19 pandemic.” 

Point#4:

Abstract: 

Objective the review was not well stated

Authors’ Response: 

Thank you for your valuable note. We incorporated the following in the abstract’s background. 

“The goal of this review is to form a preliminary and contemporary understanding of the general public knowledge, attitude, and behaviors towards the COVID-19 pandemic globally.” 

Point#5:

Abstract: 

Materials and Methods

Methods of data extraction and effect size measures were not specified.

The quality of the evidence assessment was not explained.

Authors’ Response: 

Thank you for this critical comment. We incorporated the following as part of the methods in the abstract of the paper:

“A systematic search was conducted in various databases until May 2020. Each study’s characteristics including the sample size, region, and study type were examined individually. A meta-analysis with a random-effects model and pooled prevalence with 95% confidence interval (CI) of all evaluated outcomes such as adequate knowledge, positive feelings, worrisome about the COVID-19 pandemic, and practice were recorded and reported from each study. Parameters such as random distribution, blinding, incomplete outcome data, selective reporting, and other biases were utilized to assess the quality of each retrieved record. A funnel plot was employed to assess publication bias.” 

Point#6:

Abstract: 

Results

The finding the review was not well written up.

Authors’ Response: 

Thank you for this critical comment. We incorporated the following as part of the methods in the abstract of the paper:

“A total of 26 studies with 67,143 participants were analyzed. The overall prevalence of knowledge, positive attitude, worrisome, and practice of precautionary measures were 0.87, 0.85, 0.71, and 0.77, respectively. Subgroup analysis demonstrated that social distancing is less practiced in Africa than other regions (p=0.02), while knowledge of prevention of COVID-19 was reported higher in Asia (p=0.001). Furthermore, people in developing countries had a higher prevalence of worrisome towards the COVID-19 pandemic with a p-value of less than 0.001. The funnel plot demonstrated a presence of publication bias.”

Point#7:

Introduction

Background

The research gap was not explained.

Authors’ Response: 

Similarly, as we indicated in the abstract. We highlighted the gap of knowledge by stating it in the introduction again. We incorporated the following: 

“While the current literature about the COVID-19 pandemic extensively addresses clinical and laboratory-based studies, a gap remains still present in terms of evaluating the general public knowledge and behaviors towards the COVID-19 pandemic.” 

Point#8:

Materials and methods

What are your review research development?

Authors’ Response: 

Several factors played a role in our research development. First, the risk perception of the public is very important in the overall public health response to the COVID-19 pandemic. Therefore, we touched on a topic that was not well addressed in the literature. Second, the best approach we thought about to have contemporary and preliminary evidence is by utilizing the highest level of evidence which is systematic review and meta-analysis. We also embraced the concept of the Knowledge-attitude-behavior model in the evaluation of the general public risk perception. Since there have been no interventional studies, we utilized the prevalence meta-analysis strategy which is based on pooled prevalence, as the effect size. 

Point#8:

Materials and methods

Search strategy

The authors didn’t use MeSH terms which was inclusive

What did the authors use to avoid duplication of records?

Authors’ Response: 

We utilized the Medical Subject Heading (MeSH) feature during our search strategy. Therefore, we used a wide variet of terms and keywords separate as well as in combination to be inclusive. We added a supplemntary table for the search strategy. We used EndNote to ensure de-duplication. 

Point#9:

Data extraction

Outcomes assessment was not explained

Quality and risk bias assessment

Authors’ Response: 

We have utilized the Cochrane’s review guidelines for risk of bias assessment of each included study. This model has two outcome assessment components: incomplete outcome data and selective reporting. We have reported our findings in Table 2. 

Point#10:

When the 2 author reviewers faced disagreement, how did you resolve it?

Authors’ Response: 

In the event of disagreement, the first and corresponding authors met to discuss and reach consensus. 

Point#11:

How could blinding be an evaluation item for quality assessment as your studies included was cross sectional study design?

Authors’ Response: 

Thank you for this valuable comment. True, with cross sectional study, blinding is not feasible. So, we utilized the the Cochrane’s review guidelines for quality assessment of cross sectional studies. In the absence of blinding, which was the case for all the studies, we labeled it as unclear in the quality assessment table.

 Point#12:

What did you do if certain studies didn’t meet methodological criteria?

Authors’ Response: 

We determined the inclusion criteria in the PRISMA chart in the paper. Any study that did not meet the inclusion criteria, we excluded it. However, for the methodological criteria, we did not have specific criteria as all the studies were of a cross sectional study design. We evaluated the quality according to the Cochrane’s review guidelines. 

Point#12:

Statistical analysis

How did you express your effect size?

Authors’ Response: 

Thank you for the comment. Since it was a prevalence meta-analysis, we used the prevalence as the effect size in this scenario. It was expressed as proprtions with 95% confidence interval.

Point#13:

How did you assess publication bias more objectively? Explain it by using statistical tests

Authors’ Response: 

Thank you for this critical point. We were limited in this aspect as the software we used does not offer the objective emasure of publication bias. Therefore, we evaluated it sujecictively based on the observation of the symmetry or asymmetry of the funnel plot. 

Point#14:

Quality of evidence assessment for major outcomes was not explained? Use Grade pro soft ware.

Authors’ Response: 

There are different quality assessment models that have been employed to evaluate the quality of evidence of cross-sectional studies. The authors have utilized the Cochrane’s review guidelines to evaluate the quality of each reported study. Thank you for your suggestion to use GradePro, however, up to the knowledge of authors, we were not able to find meta-analysis that has utilized GradePro on cross-sectional studies. Of note, below is another meta-analysis that has utilized the Cochrane’s review guidelines for risk of bias assessment of cross-sectional studies. 

Guo S, Yang Y, Liu, F, Li F. The awareness rate of mental health knowledge Among Chinese adolescent. Medicine 2020; 99(7). 

Point#15:

Results

The pooled prevalence of each outcome along with CI should be specified separately in a logical order

Authors’ Response: 

We have added another table (Table 3) in the results section to depict pooled prevalence and confidence interval of major outcomes in logical order. 

Point#15:

Discussion

The finding was not compared and contrasted.

The possible reason of the differences/similarities should be stated

Authors’ Response: 

Thank you for this comment. We performed compare and contrast among individual studies. Due to lack of evidence in this suject especially that our report is the first to illuminate prevalence meta analysis among the general public, we however, compared and contrasted to previous studies with similar outbreaks. We provided several explanations for the differences and smiliarties to the best of our knowledge and interpretation of the existing literature. 

Point#15:

Conclusion

It should be made in line to the findings.

Authors’ Response: 

Thank you. Changed per recommendation.

---

## [Decision Letter · Decision Letter 1]

23 Dec 2020

PONE-D-20-19020R1

The current global perspective of the knowledge-attitude-behavior of the general public towards the corona virus disease -19 pandemic: Systematic review and meta-analysis on 67,143 participants.

PLOS ONE

Dear Dr. Shareef,

Thank you for submitting your manuscript to PLOS ONE. After careful consideration, we feel that it has merit but does not fully meet PLOS ONE’s publication criteria as it currently stands. Therefore, we invite you to submit a revised version of the manuscript that addresses the points raised during the review process.

Dear Authors, thank you very much for responding to the queries of referees. There are few more comments need your attention. Publication Bias and Quality assessment is area of concerns still needed to be addressed.

We look forward to receiving your revised manuscript.

Kind regards,

Tauqeer Hussain Mallhi, Ph.D

Academic Editor

PLOS ONE

Reviewers' comments:

Reviewer's Responses to Questions

**Comments to the Author**

1. If the authors have adequately addressed your comments raised in a previous round of review and you feel that this manuscript is now acceptable for publication, you may indicate that here to bypass the “Comments to the Author” section, enter your conflict of interest statement in the “Confidential to Editor” section, and submit your "Accept" recommendation.

Reviewer #1: All comments have been addressed

Reviewer #2: All comments have been addressed

Reviewer #3: (No Response)

2. Is the manuscript technically sound, and do the data support the conclusions?

Reviewer #1: Yes

Reviewer #2: Yes

Reviewer #3: Yes

3. Has the statistical analysis been performed appropriately and rigorously? 

Reviewer #1: Yes

Reviewer #2: Yes

Reviewer #3: Yes

4. Have the authors made all data underlying the findings in their manuscript fully available?

Reviewer #1: Yes

Reviewer #2: Yes

Reviewer #3: Yes

5. Is the manuscript presented in an intelligible fashion and written in standard English?

Reviewer #1: Yes

Reviewer #2: Yes

Reviewer #3: No

6. Review Comments to the Author

Reviewer #1: The authors have done a good job. They have addressed all the technical and grammatical errors in the manuscript. They have also addressed all the comments that i raised. The manuscript is technically sound and should be accepted for publication in you journal.

Reviewer #2: Thank you for successfully addressed all the points. The manuscript is now suitable for publication in the journal.

Reviewer #3: I have scanned the authors' work meticulously. The authors have tried to react to certain questions; nevertheless, publication bias and quality evidence assessment issues has not been addressed yet. Hint : Publication bias can be assessed objectively using statistical tests like Beggs test, Egger test and harbord test using Stata version 16/ its extension or comprehensive meta analysis software whereas quality evidence assessment can be made using Grade pro soft ware. The authors are suggested to down load the aforementioned softwares and use it.

7. PLOS authors have the option to publish the peer review history of their article (what does this mean?). If published, this will include your full peer review and any attached files.

Reviewer #1: No

Reviewer #2: No

Reviewer #3: No

---

## [Author Response · Author response to Decision Letter 1]

7 Feb 2021

We have performed quality evidence assessment of each reported outcome and used Grade pro software for that as suggested. In addition, both Begg’s and Egger’s tests were used to evaluate symmetry of funnel plots for assessment of publication bias. We have reported all the findings in the revised manuscript. We have utilized MedCalc software for meta-analysis per authors’ convenience. Thank you for your valuable suggestion.

---

## [Decision Letter · Decision Letter 2]

28 Mar 2021

PONE-D-20-19020R2

The current global perspective of the knowledge-attitude-behavior of the general public towards the corona virus disease -19 pandemic: Systematic review and meta-analysis on 67,143 participants.

PLOS ONE

Dear Dr. Shareef,

Thank you for submitting your manuscript to PLOS ONE. After careful consideration, we feel that it has merit but does not fully meet PLOS ONE’s publication criteria as it currently stands. Therefore, we invite you to submit a revised version of the manuscript that addresses the points raised during the review process.

We look forward to receiving your revised manuscript.

Kind regards,

Tauqeer Hussain Mallhi, Ph.D

Academic Editor

PLOS ONE

Journal Requirements:

Additional Editor Comments (if provided):

Thank you very much for revisions. However, reviewer has raised few more concerns. Please address the reviewer`s queries so appropriate decision could be made.

Reviewers' comments:

Reviewer's Responses to Questions

**Comments to the Author**

1. If the authors have adequately addressed your comments raised in a previous round of review and you feel that this manuscript is now acceptable for publication, you may indicate that here to bypass the “Comments to the Author” section, enter your conflict of interest statement in the “Confidential to Editor” section, and submit your "Accept" recommendation.

Reviewer #1: All comments have been addressed

Reviewer #3: All comments have been addressed

2. Is the manuscript technically sound, and do the data support the conclusions?

Reviewer #1: (No Response)

Reviewer #3: Yes

3. Has the statistical analysis been performed appropriately and rigorously? 

Reviewer #1: Yes

Reviewer #3: Yes

4. Have the authors made all data underlying the findings in their manuscript fully available?

Reviewer #1: Yes

Reviewer #3: Yes

5. Is the manuscript presented in an intelligible fashion and written in standard English?

Reviewer #1: Yes

Reviewer #3: Yes

6. Review Comments to the Author

Reviewer #1: (No Response)

Reviewer #3: I have meticulously scanned the authors manuscript. My concerns were addressed. Nevertheless, there was some concerns. eg in abstract : background : the authors should state his purpose by using the aim instead of using goal,

Materials and methods in main text:

spelling error should be corrected: review manager not review manage

7. PLOS authors have the option to publish the peer review history of their article (what does this mean?). If published, this will include your full peer review and any attached files.

Reviewer #1: No

Reviewer #3: No

---

## [Author Response · Author response to Decision Letter 2]

29 Mar 2021

As per the third reviewer’s suggestion, we have replaced the word “goal” with “aim” in the abstract and we have corrected the spelling of “review manager” in the methods section. Thank you for your valuable feedback.

---

## [Decision Letter · Decision Letter 3]

27 May 2021

PONE-D-20-19020R3

The current global perspective of the knowledge-attitude-behavior of the general public towards the corona virus disease -19 pandemic: Systematic review and meta-analysis on 67,143 participants.

PLOS ONE

Dear Dr. Shareef,

Thank you for submitting your manuscript to PLOS ONE. After careful consideration, we feel that it has merit but does not fully meet PLOS ONE’s publication criteria as it currently stands. Therefore, we invite you to submit a revised version of the manuscript that addresses the points raised during the review process.

We look forward to receiving your revised manuscript.

Kind regards,

Tauqeer Hussain Mallhi, Ph.D

Academic Editor

PLOS ONE

Journal Requirements:

Additional Editor Comments (if provided):

Thank you very much for revising the draft. I invite you to consider the few more comments from the one reviewer.

Reviewers' comments:

Reviewer's Responses to Questions

**Comments to the Author**

1. If the authors have adequately addressed your comments raised in a previous round of review and you feel that this manuscript is now acceptable for publication, you may indicate that here to bypass the “Comments to the Author” section, enter your conflict of interest statement in the “Confidential to Editor” section, and submit your "Accept" recommendation.

Reviewer #1: All comments have been addressed

Reviewer #3: (No Response)

2. Is the manuscript technically sound, and do the data support the conclusions?

Reviewer #1: Yes

Reviewer #3: Yes

3. Has the statistical analysis been performed appropriately and rigorously? 

Reviewer #1: Yes

Reviewer #3: Yes

4. Have the authors made all data underlying the findings in their manuscript fully available?

Reviewer #1: Yes

Reviewer #3: Yes

5. Is the manuscript presented in an intelligible fashion and written in standard English?

Reviewer #1: Yes

Reviewer #3: Yes

6. Review Comments to the Author

Reviewer #1: (No Response)

Reviewer #3: The overall prevalence of knowledge, positive attitude, worrisome, and practice of precautionary

measures should be indicated along with 95% confidence interval both in the abstract and main body part of the results.

7. PLOS authors have the option to publish the peer review history of their article (what does this mean?). If published, this will include your full peer review and any attached files.

Reviewer #1: No

Reviewer #3: No

---

## [Author Response · Author response to Decision Letter 3]

27 Jun 2021

Reviewers' Comments to the Authors:

Reviewer #3: The overall prevalence of knowledge, positive attitude, worrisome, and practice of precautionary measures should be indicated along with 95% confidence interval both in the abstract and main body part of the results.

Authors’ Response: 

As per the third reviewer’s suggestion, we ensured that all pooled prevalence are associated with 95% confidence interval in both abstract and results sections. Thank you for your comment.

---

## [Decision Letter · Decision Letter 4]

21 Jul 2021

PONE-D-20-19020R4

The current global perspective of the knowledge-attitude-behavior of the general public towards the corona virus disease -19 pandemic: Systematic review and meta-analysis on 67,143 participants.

PLOS ONE

Dear Dr. Shareef,

Thank you for submitting your manuscript to PLOS ONE. After careful consideration, we feel that it has merit but does not fully meet PLOS ONE’s publication criteria as it currently stands. Therefore, we invite you to submit a revised version of the manuscript that addresses the points raised during the review process.

We look forward to receiving your revised manuscript.

Kind regards,

Tauqeer Hussain Mallhi, Ph.D

Academic Editor

PLOS ONE

Additional Editor Comments (if provided):

Thank you for revising the draft. This manuscript has been substantially improved. However, few changes are suggested by the reviewer. Moreover, It is advised to take assistance from native English speaker to revise the draft for Grammar, Syntax and spelling.

Journal Requirements:

Reviewers' comments:

Reviewer's Responses to Questions

**Comments to the Author**

1. If the authors have adequately addressed your comments raised in a previous round of review and you feel that this manuscript is now acceptable for publication, you may indicate that here to bypass the “Comments to the Author” section, enter your conflict of interest statement in the “Confidential to Editor” section, and submit your "Accept" recommendation.

Reviewer #1: All comments have been addressed

Reviewer #3: (No Response)

2. Is the manuscript technically sound, and do the data support the conclusions?

Reviewer #1: Yes

Reviewer #3: Yes

3. Has the statistical analysis been performed appropriately and rigorously? 

Reviewer #1: Yes

Reviewer #3: No

4. Have the authors made all data underlying the findings in their manuscript fully available?

Reviewer #1: Yes

Reviewer #3: No

5. Is the manuscript presented in an intelligible fashion and written in standard English?

Reviewer #1: Yes

Reviewer #3: Yes

6. Review Comments to the Author

Reviewer #1: (No Response)

Reviewer #3: Abstract

Background

There was tense error eg

The aim of this review is to form a preliminary and contemporary understanding of the general public knowledge, attitude, and behaviors towards the COVID-19 pandemic globall

Results

The overall prevalence of knowledge, positive attitude, worrisome, and practice of precautionary

measures can't be computed using RevMan software. Use other POWERFUL software like STATA or other software.

7. PLOS authors have the option to publish the peer review history of their article (what does this mean?). If published, this will include your full peer review and any attached files.

Reviewer #1: No

Reviewer #3: No

---

## [Author Response · Author response to Decision Letter 4]

19 Sep 2021

Comments of Editor:

Editor: Thank you for revising the draft. This manuscript has been substantially improved. However, few changes are suggested by the reviewer. Moreover, It is advised to take assistance from native English speaker to revise the draft for Grammar, Syntax and spelling.

Authors’ Response: 

We have initially consulted with a professional medical writing service that corrected the mechanical errors and performed linguistic editing in the manuscript. The manuscript was revised again and grammatical errors were corrected. 

Reviewers' Comments to the Authors:

Reviewer #3: Abstract

Background

There was tense error eg

The aim of this review is to form a preliminary and contemporary understanding of the general public knowledge, attitude, and behaviors towards the COVID-19 pandemic global

Authors’ Response: 

The manuscript was revised for any tense error and corrected accordingly. All changes have been implemented in the final manuscript. Thanks for your suggestion. 

Reviewer #3:

Results

The overall prevalence of knowledge, positive attitude, worrisome, and practice of precautionary measures can't be computed using RevMan software. Use other POWERFUL software like STATA or other software.

Authors’ Response: 

We have calculated pooled prevalence of studies using RevMan Software which computes the prevalence as the effect size along with 95% confidence interval. Since RevMan has an option of calculating pooled prevalence, we decided to stick with this software. Thank you for the suggestion.

---

## [Decision Letter · Decision Letter 5]

8 Nov 2021

The current global perspective of the knowledge-attitude-behavior of the general public towards the corona virus disease -19 pandemic: Systematic review and meta-analysis on 67,143 participants.

PONE-D-20-19020R5

Dear Dr. Shareef,

We’re pleased to inform you that your manuscript has been judged scientifically suitable for publication and will be formally accepted for publication once it meets all outstanding technical requirements.

Kind regards,

Tauqeer Hussain Mallhi, Ph.D

Academic Editor

PLOS ONE

Additional Editor Comments (optional):

Reviewers' comments:

Reviewer's Responses to Questions

**Comments to the Author**

1. If the authors have adequately addressed your comments raised in a previous round of review and you feel that this manuscript is now acceptable for publication, you may indicate that here to bypass the “Comments to the Author” section, enter your conflict of interest statement in the “Confidential to Editor” section, and submit your "Accept" recommendation.

Reviewer #3: All comments have been addressed

2. Is the manuscript technically sound, and do the data support the conclusions?

Reviewer #3: Yes

3. Has the statistical analysis been performed appropriately and rigorously? 

Reviewer #3: Yes

4. Have the authors made all data underlying the findings in their manuscript fully available?

Reviewer #3: Yes

5. Is the manuscript presented in an intelligible fashion and written in standard English?

Reviewer #3: Yes

6. Review Comments to the Author

Reviewer #3: The authors have substantially improved their manuscript.

Title:

The number indicated on title "on 67,143 participants'' should be expunged.

Abstract

The authors should follow the PLOS ONE guidelines

Use unstructured abstract

The first 'COVID 19 " should be non abbreviated; subsequently, the authors can use abbreviation.

Main body

methods

The authors should state inclusion criteria (PCC), and study selection.

Data synthesis

Specify the cut off to deem presence of heterogeneity and publication bias.

correct language error

Rewrite" The outcome data are presented as mean with a 95% confidence interval.

7. PLOS authors have the option to publish the peer review history of their article (what does this mean?). If published, this will include your full peer review and any attached files.

Reviewer #3: No

---

## [Editor Report · Acceptance letter]

15 Nov 2021

PONE-D-20-19020R5 

The current global perspective of the knowledge-attitude-behavior of the general public towards the corona virus disease -19 pandemic: Systematic review and meta-analysis on 67,143 participants. 

Dear Dr. Shareef:

I'm pleased to inform you that your manuscript has been deemed suitable for publication in PLOS ONE. Congratulations! Your manuscript is now with our production department. 

Kind regards, 

on behalf of

Dr. Tauqeer Hussain Mallhi 

Academic Editor

PLOS ONE